# Tens of thousands additional deaths annually in cities of China between 1.5 °C and 2.0 °C warming

Yanjun Wang [1], Anqian Wang [2,3], Jianqing Zhai[4], Hui Tao [2], Tong Jiang [1], Buda Su [2], Jun Yang[5], Guojie Wang[1], Qiyong Liu[6], Chao Gao [7], Zbigniew W. Kundzewicz[1,8], Mingjin Zhan [9], Zhiqiang Feng[10] & Thomas Fischer [11]

The increase in surface air temperature in China has been faster than the global rate, and more high temperature spells are expected to occur in future. Here we assess the annual heat-related mortality in densely populated cities of China at 1.5 °C and 2.0 °C global warming. For this, the urban population is projected under five SSPs, and 31 GCM runs as well as temperature-mortality relation curves are applied. The annual heat-related mortality is projected to increase from 32.1 per million inhabitants annually in 1986–2005 to 48.8–67.1 per million for the 1.5 °C warming and to 59.2–81.3 per million for the 2.0 °C warming, taking improved adaptation capacity into account. Without improved adaptation capacity, heat-related mortality will increase even stronger. If all 831 million urban inhabitants in China are considered, the additional warming from 1.5 °C to 2 °C will lead to more than 27.9 thousand additional heat-related deaths, annually.

[1] Institute for Disaster Risk Management /School of Geographical Science, Nanjing University of Information Science & Technology, Nanjing 210044, China. [2] State Key Laboratory of Desert and Oasis Ecology, Xinjiang Institute of Ecology and Geography, Chinese Academy of Sciences, Urumqi 830011, China. [3] University of Chinese Academy of Sciences, Beijing 100049, China. [4] National Climate Center, China Meteorological Administration, Beijing 100081, China. [5] Institute for Environmental and Climate Research, Jinan University, Guangzhou 511443, China. [6] National Institute for Communicable Disease Control and Prevention, Chinese Center for Disease Control and Prevention, Beijing 102206, China. [7] Faculty of Architectural, Civil Engineering and Environment, Ningbo University, Ningbo 31511, China. [8] Institute for Agricultural and Forest Environment, Polish Academy of Sciences, Poznan, Poland. [9] Jiangxi Climate Center, Nanchang 330096, China. [10] School of Geosciences, University of Edinburgh, Edinburgh EH8 9XP, UK. [11] Department of Geosciences, Eberhard Karls University, Tübingen 72070, Germany. Correspondence and requests for materials should be addressed to T.J. (email: jiangtong@cma.gov.cn) or to B.S. (email: subd@cma.gov.cn) or to T.F. (email: thomas.fischer.geo@gmx.de)

Climate change is the biggest global threat of the 21st century[1]. Adverse weather events are projected to increase dramatically in frequency, severity, and duration. Global warming is projected to affect human health, with primarily negative consequences of increasing number of excess deaths and hospital admission worldwide[2–5]. In the recent past, numerous extreme high temperature events with associated mortality have taken place worldwide. For instance, the heat wave of 2003 in Europe resulted in more than 70,000 additional deaths[6,7]. An unprecedented high temperature event in Moscow and Western Russia in the summer of 2010 led to nearly 55,000 excess deaths[8,9]. A record-breaking high temperature event in Shanghai, China in 2013 brought 160 excess deaths in Pudong New District alone[10]. Considering ever worsening situation, it is of utmost importance to project the adverse health effects of high temperature to support the developing of targeted intervention strategies for public health protection.

Impacts of future climate extremes on public health have been a major research topic in recent years[5,11–15]. The Special Report on Global Warming of 1.5 °C emphasized that, with high confidence, an increase in heat-related mortality caused by high temperature at 1.5 °C and 2.0 °C threshold levels is apparent[16]. Although the decrease in cold season low-temperature extremes is expected to result in lower mortality rates during the winter months, the increase in heat-related mortality could outweigh such reductions in cold-related mortality, even in regions with colder climate[3,17,18]. Studies have consistently projected that a warmer future will lead to increases in future mortality with tens of thousands of additional premature deaths per year in the United States, and over a hundred thousand per year globally[19,20]. Still, projecting changes in future health impacts associated with climate warming remains challenging and involves large uncertainties. In particular, little is known about future impacts of heat waves in less developed countries, where capacity to address climate change is comparatively low and vulnerability to climate-related damages is high.

Most projections of heat-related mortality under climate change did not account for population acclimatization to heat stress. People may adapt to heat stress through modifications in activities, increased use of air conditioning, and alternative building designs[21]. Projecting future mortality effects of climate change without considering human adaptability may lead to a substantial overestimation[22,23]. On the other hand, due to differences of the gender- and age-related physiological and thermoregulatory properties, increase in vulnerable population may amplify future heat-related health impacts. The fact that changes in these demographic structures have not been considered in previous studies may have caused an underestimation of mortality due to climate change[5,14,24–27].

China is the largest developing country, and has a faster increase in surface air temperature than the global average[20,28,29]. The elderly population is increasing and will continue to increase further in the 21st century even after the end of the one-child policy. As a result, the heat-related health risk will probably be aggravated in future. However, only a few studies focused on heat-related health impacts in China[11,14,20,27,30,31], and they often ignored the changing population structure and adaptation capacity. In our study, the heat-related mortality in major cities of China is assessed by applying case analyses from 27 metropolises (Supplementary Fig. 1 and Supplementary Table 1) for 1.5 °C and 2.0 °C global warming. The mortality projections are based on an integrated assessment framework that combines projected high temperature from multiple GCMs, predicted population by gender and age structure under five SSPs, and a dynamic temperature-mortality relationship with consideration of improving adaptation capacity. In addition to the changes in the mortality-inducing high temperature, the differences of mortality between various climate and socioeconomic scenarios are also assessed to deepen our understanding of the potential benefits of climate change mitigation that will limit global warming.

## Results

**Definition of threshold temperature.** Global mean surface air temperature of 1986–2005 was by 0.61 °C warmer than the pre-industrial level[32], and further increase to 0.87 °C (likely between 0.75 °C and 0.99 °C) for the decade 2006–2015 was reported[16]. The ensemble mean of 31 GCM outputs (Supplementary Fig. 2 and Supplementary Table 2) of the Coupled Model Inter-comparison Project phase 5 (CMIP5) shows that a 20-year moving average of global mean temperature may reach 1.5 °C global warming around 2030 under RCP2.6, and 2.0 °C around 2050 under RCP4.5. The projected temperature shows a low variation after the 2060s under both pathways[33–35]. In order to conduct an impact study under comparative stable climatic conditions, we choose the time period of 1986–2005 as the reference period and the future time horizon of 2060–2099 under RCP2.6 for 1.5 °C global warming and under RCP4.5 for 2.0 °C global warming, although there will be overshoot.

Existing studies identified a non-linear U-, V- or J-shaped relationship between temperature and mortality, suggesting that the mortality will sharply increase once a certain threshold is exceeded[5,36–40]. We classified all heat-related mortality cases of 27 metropolises during the time period 2007–2013 into four groups by gender (male and female) and age (working age: 15–64 years and non-working age: ≤14 and ≥65 years). In the follow-up, a distributed lag non-linear model (DLNM) was applied to identify the temperature-mortality relationship for each group. The DLNM model is used to estimate the relative risk (RR) of mortality for each temperature, and RR = 1 corresponding to the mortality-inducing threshold temperature (see "Methods", Supplementary Fig. 3 and Supplementary Table 3). Once daily maximum temperature reaches or exceeds the threshold, these days are counted as days with high temperature. The intensity of high temperature is defined as the range of temperature (in degrees Celsius) over the threshold.

**Trends in high temperature.** Temperature thresholds of mortality vary for different gender and age groups. The lowest threshold corresponding to mortality-inducing temperature for female non-working age population was selected to assess the changes of frequency and intensity of high temperature in each China metropolis. According to the ensemble mean of 31 GCM outputs, annual frequency of high temperature averaged over 27 metropolises shows a significant positive trend of 1.5d/10a during 1961–2005, and continuously, a significant upward trend is projected until the 2050s. The rate of the increase will go to zero (RCP2.6) or slow down (RCP4.5) after the 2050s. With global warming of 1.5 °C or 2.0 °C, on average, 67.1 or 73.8 days of high (mortality-inducing) temperature, respectively, will occur per year in 2060–2099. This is an increase by 32.6% or 45.8%, respectively, relative to 50.6 days during 1986–2005 (Fig. 1a).

The annual mean intensity of high temperature during 1961–2005 shows an increasing trend of 0.07 °C/10a. Similar to the frequency, the intensity will increase continuously until the 2050s under both pathways, RCP2.6 and RCP4.5. After the 2050s, the intensity will not increase under RCP2.6, but will still increase under RCP4.5. The intensity in the reference period was approximately equal to 1.6 °C. Compared with the reference period, the intensity of high temperature is projected to increase by 1.2 °C and 1.9 °C at a global warming of 1.5 °C and 2.0 °C, respectively (Fig. 1b).

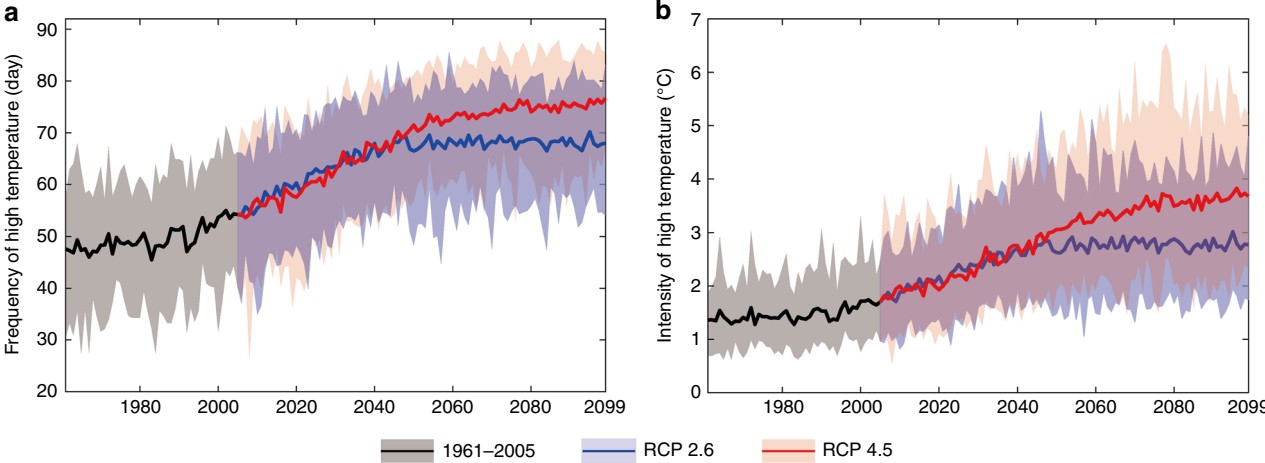

**Fig. 1** Frequency and intensity of high temperature in China metropolises for 1961–2099. Curves and shadows denote ensemble mean and range of 31 GCMs, respectively. Frequency (**a**) and intensity (**b**) of high temperature in China metropolises for the reference period 1961–2005 (gray) and for the future period 2005–2099 with RCP 2.6 (blue) and RCP 4.5 (red). Curves and shadows denote the ensemble mean and range of 31 GCMs, respectively. Source data are provided as a Source Data file

**Changes in total mortality**. As changing exposure and improved adaptation capacity change the risks of climate extremes, an adequate assessment of climate change impacts should take future socioeconomic development into account. Therefore, the population by age and gender, and the Gross Domestic Product (GDP) of 27 metropolises in China for the 21st century are projected under the framework of the Shared Socioeconomic Pathways (SSPs), which represent different climate strategies for mitigation and adaptation (Supplementary Fig. 4 and Supplementary Table 4). The SSPs describe a set of plausible alternative futures of societal development, which consider the effects of climate change and new climate policies. The SSPs include a pathway of a sustainable world (SSP1), a pathway of continuing historical trend (SSP2), a strongly fragmented world (SSP3), a highly unequal world (SSP4), and a growth-oriented world (SSP5)[41,42]. All five SSPs combined with RCP2.6 and RCP4.5 can produce ten plausible climatic-socioeconomic scenarios for the assessment of risks from high temperature. Additionally, GDP per capita in metropolises can be used as an indicator to evaluate the adaptability of different cities to high temperature (Supplementary Fig. 5).

On average, heat-related mortality in China metropolises was 32.1 per million by ensemble mean of the multiple GCMs in 1986–2005 (Fig. 2). Under the assumption that the socio-economy remains stable at the 1986–2005 status, increasing frequency and intensity of high temperature will double the heat-related mortality to 64.3 per million at global warming of 1.5 °C, and even stronger increase to 85.5 per million at 2.0 °C global warming (Supplementary Table 5).

However, exposure and vulnerability to high temperature are dynamic, and human adaptability to adverse climate is expected to increase with the socioeconomic development. When improved adaptation is integrated into assessment, interaction between the severity of high temperature and an increase in vulnerable population in the future will lead to increases in heat-related mortality to 48.8–67.1 per million for 1.5 °C global warming, across plausible development pathways, and to 59.2–81.3 per million for 2.0 °C global warming (Fig. 2). That is to say, curbing the increase in global temperature to 1.5 °C can reduce heat-related mortality in China metropolises by about 18% compared with 2.0 °C.

Ignorance of contribution of adaptation actions could lead to substantial overestimation of climate change impacts. Without improved adaptation, heat-related mortality will be enlarged to 103.7–129.9 per million for 1.5 °C global warming under various SSPs. Further increase in mortality to 137.3–169.9 per million was projected for 2.0 °C warming (Fig. 2). For the urban population of 831 million in China, the extra heat-related mortality between 1.5 °C and 2.0 °C global warming will be in the range of 27.9–33.2 thousands, annually.

**Changes in gender- and age-specific mortality**. The heat-related mortality in China metropolises in 1986–2005 is equal to 22.0 female and 10.1 male cases per million. Under various SSPs at 1.5 °C global warming, mortality will increase to 30.3–40.9 per million (relative increase of 37.7%–85.9%) for the female population and even faster (by 83.2%–160.4% to 18.5–26.3 per million) for the male population. At 2.0 °C global warming, mortality in female population will increase by 61.4%–118.2% to 35.5–48.0 per million, and of the male population will increase by 134.7%–229.7% to 23.7–33.3 per million (Fig. 3a and Supplementary Table 6). Overall, female mortality was and will be continuously higher than for male, but the gap between genders is projected to be narrowed, due to the assumed changes in sex ratio in China from 105:100 in 1986–2005, for various SSPs, to (96–101):100 in 2060–2099.

If no improvement in adaptation capacity is assumed, mortality in the female and male population will be 71.2–88.0 and 32.4–42.0 per million, respectively, at 1.5 °C global warming, and will further increase to 93.9–114.4 and 43.4–55.4 per million, respectively, at 2.0 °C global warming. Improved adaptability can reduce 36.8%–43.0% of mortality in the male population and 52.8%–57.5% of the female population at 1.5 °C global warming, while it reduces 39.3%–45.5% of mortality in the male population, and 57.2%–62.2% of the female population at 2.0 °C global warming (Supplementary Fig. 6a).

For 1986–2005, heat-related mortality in the working age population was 7.0 per million and that of the non-working age population was 25.1 per million. With 1.5 °C global warming, mortality in the working age population is projected to decrease significantly by 42.9%–60.0% to 2.8–4.1 per million. In contrast, mortality in the non-working age population is projected to increase significantly to 44.7–64.4 per million. This is an increase by 78.1%–156.6% compared to the reference period. With 2.0 °C global warming, the mortality in the working age population will

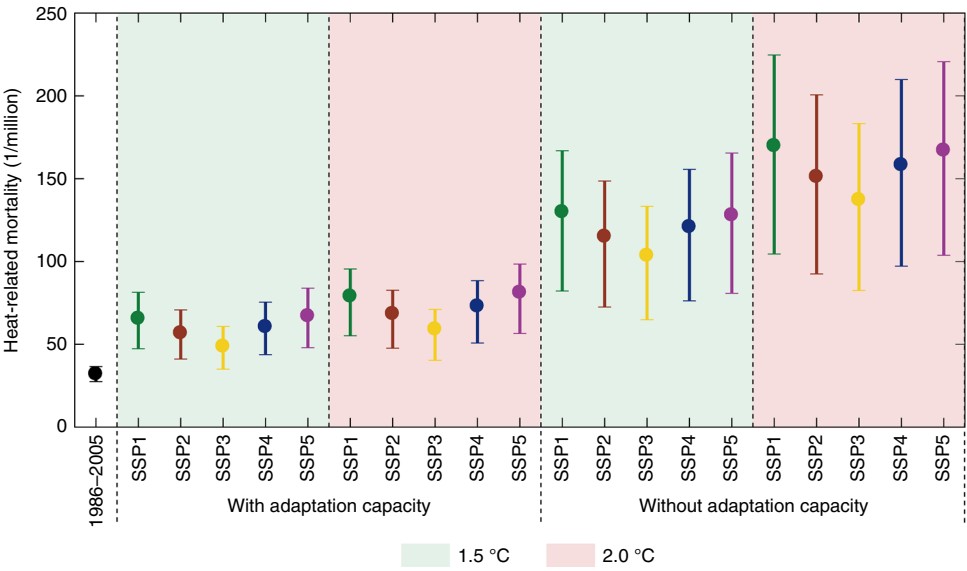

**Fig. 2** Comparison of annual heat-related mortality at 1.5 °C and 2.0 °C global warming under SSPs and the reference period (1986–2005). Future projection of mortality considers two scenarios—with and without improved adaptation capacity. Dots and straight lines denote the ensemble mean and range of mortality estimated by multiple GCMs. Source data are provided as a Source Data file

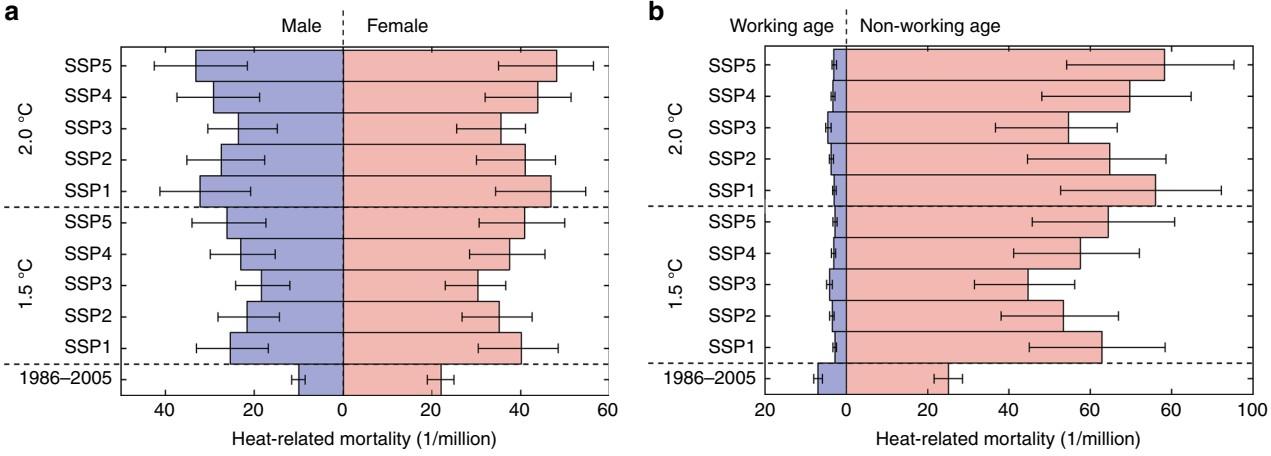

**Fig. 3** Comparison of annual gender and age-specific heat-related mortality at 1.5 °C and 2.0 °C global warming under SSPs and the reference period (1986–2005). Comparison of annual gender (**a**) and age (**b**) specific heat-related mortality at 1.5 °C and 2.0 °C global warming under SSPs and the reference period (1986–2005). Colored bars and black straight lines denote the ensemble mean and range of mortality estimated by multiple GCMs. Source data are provided as a Source Data file

significantly decrease by 35.7%–57.1% to 3.0–4.5 per million. As for the non-working age population, it will significantly increase by 117.5%–211.6% to 54.7–78.2 per million. The increase of heat-related mortality for the non-working age population and decrease for the working age population in China metropolises with the warming are mainly due to the projected demographic structure changes (Fig. 3b and Supplementary Table 6).

Under scenario without improved adaptation capacity, mortality will be 162.5%–167.9% higher for the working age population, and 87.1%–108.5% higher for the non-working age population than projections with improved adaptability, at 1.5 °C global warming. Mortality will be 224.4%–240.0% higher for the working age population and 100.6%–124.7% higher for the non-working age population, with the additional increase in global warming by 0.5 °C (Supplementary Fig. 6b).

## Discussion
With global warming, temperature extremes are likely to be more frequent, more intense, and longer lasting. In addition, demographics and adaptation capacities will change dramatically in future. The assessment of future changes in heat-related mortality requires projections of the climate conditions, the population growth, the socioeconomic development, and consideration of improved adaptation. As far as we are aware, this is the first attempt to use locally defined concepts to investigate the relationship between high temperature and mortality for a large fraction of major cities in China. In this study, recorded cases from 27 metropolises are applied to deduce the threshold temperature for heat-related mortality. Furthermore, daily maximum temperature from 31 GCM outputs are combined with projected population under five SSPs to estimate mortality at 1.5 °C and 2.0 °C global warming,

by considering improved adaptation capacities under various economic development scenarios.

Heat-related mortality increases above a certain threshold temperature with a non-linear relationship. This threshold temperature is the most critical information in preventing the health impacts of high temperature, as it is an indicator for initiating public health responses[5,43,44]. The threshold temperature is the temperature at which adverse health effects from heat begin to occur. The impacts are diverse for various categories, e.g. gender and age groups or geography[45]. Kan et al. investigated the relationship between daily mean temperature and mortality in Shanghai from January 2000 to December 2001 by using a generalized additive model, and found a gently sloping V-like relationship with the lowest mortality risk temperature of 26.7 °C[46]. Another heat-related mortality study by Knowlton et al. found that the threshold temperature in New York is ~23.1 °C[13]. In our study, the gender- and age-specific mortality-inducing threshold temperature in Shanghai ranges around 29.7–31.4 °C. In Beijing, which is located almost at the same latitude as New York, the threshold temperature is about 25.9–27.6 °C.

Direct comparisons of the impact estimations are biased as different climate models, scenarios, downscaling methods, time periods, and population growth scenarios are used. For example, the increase in heat-related mortality in Jiangsu province of eastern China was projected to reach 102 per million under RCP4.5 for 2041–2065, relative to 1981–2005[11]. An increase in mortality by 134 per million in New York and 107 per million in Philadelphia was found by Petkova et al., who used RCP4.5 scenario for 2070–2099 relative to 1971–2000 for their projections[23]. A study for 209 cities in the United States suggests that heat-related mortality increase by about 44.3 per million under RCP6.0 in 2086–2100 relative to 1976–2005[47]. To allow a rough comparison between this study and previous studies, we computed the changes in future heat-related mortality per million for scenarios not including improved adaptation capacity (Fig. 2 and Supplementary Fig. 6). Our findings of increases in future heat-related mortality are broadly consistent with these assessments. We deduced an annual heat-related mortality of 32.1 per million in the reference period. No improvement in adaptation capacity is considered, the range of heat-related mortality will be 103.7–129.9 per million at 1.5 °C global warming, and 137.3–169.9 per million at 2.0 °C global warming. The mortality in China metropolises projected in our study is higher than in the United States for the last forty years of the 21st century, which indicates a lower adaptation capacity in China than in the United States. Of course, other factors, such as the uncertainties in climate models, emission scenarios as well as baseline mortality rates, are also contributing to the differences in mortality estimations.

By incorporating future assumptions for an improved adaptability into assessment, a much lesser increase of mortality will be projected. Under improved adaptation capacity, annual heat-related mortality is projected to be 48.8–67.1 per million at 1.5 °C global warming, and 59.2–81.3 per million at 2.0 °C global warming. That is to say, improved adaptation capacity will lead to 48.3–52.9% less mortality at 1.5 °C, and 52.1–56.9% less mortality at 2.0 °C global warming. Comparing with 2.0 °C global warming, some 18% of mortality can be reduced in China metropolises by curbing temperature to 1.5 °C.

It is a common assumption that heat-related mortality is more marked in the elderly and the female population, who are more vulnerable to the impact of high temperature than the adult and male population[36,48]. Some studies highlighted that females are at higher risks of dying or being sick during high temperature episodes[45,49]. According to the relative risk of specific temperature estimated by a distributed lag non-linear model, it is found

that the threshold temperature for males is ~0.8 °C higher than for females, and for the working age population, it is 1.5 °C higher than for the non-working age population (Supplementary Table 3). With the warming, China will face adverse impacts due to the aging population. Our findings also suggest that heat-related female mortality is much higher than for males at both global warming levels, but the gap between the mortality rates in males and females will slightly narrow in future, due to changes in the sex ratio in China.

The split of the working and non-working age population is projected to change quite seriously from 75.9%:24.1% in 1986–2005 to 43.8%:56.2% in 2060–2099. As the population structure will be extremely altered, the age-specific heat-related mortality will be different at 1.5 °C global warming than at 2.0 °C global warming. At 1.5 °C global warming, the mortality in the working age population will be reduced by 42.9–60.0% relative to the reference period. On the contrary, the mortality in the non-working age population will increase significantly by 78.1–156.6%. At 2.0 °C global warming, the mortality in the working age population will be slightly higher than for 1.5 °C global warming, while for the non-working age population mortality will be much higher with 2.0 °C compared to 1.5 °C.

## Methods

**Study area**. In total, 27 major cities of China, i.e. metropolises, which include four municipalities (Beijing, Tianjin, Shanghai, and Chongqing) and most of the provincial capitals, are selected to project heat-related mortality under future climatic and socioeconomic scenarios. The population in each metropolis is above 2.0 million, and exceeds 10.0 million in Beijing, Chengdu, Chongqing, Guangzhou, Harbin, Shanghai, Shijiazhuang, and Tianjin. The total population and GDP of the 27 major cities were about 247.6 million people and 13.0 trillion CNY in 2010, which account for 18.6 and 29.7% of the national total, respectively (Supplementary Fig. 1 and Supplementary Table 1).

**Mortality records**. The daily mortality data in China metropolises during 2007–2013 were collected from the Chinese National Center for Chronic and Non-communicable Disease Control and Prevention. The underlying cause of death was coded based on the 10th Revision of the International Statistical Classification of Diseases and Related Health Problems (ICD-10). Amongst, daily non-accidental mortality (ICD-10: A00–R99), mortality due to cardiovascular disease (I00–I99), respiratory disease (J00–J99), and so on were further categorized into four groups by age and gender: working age (age: 15–64 years) and non-working age (age: ≤14 and ≥65 years); female and male. Details of the mortality data can be found in a previous study by Yang et al.[27].

**Observed and simulated climate data**. Ground-based, quality controlled, daily maximum temperature observation records in 27 China metropolises during 1961–2017 were provided by the National Climate Center of China Meteorological Administration.

The daily maximum temperature derived from 15 GCMs (CNRM-CM5, CanESM2, GFDL-CM3, GFDL-ESM2G, GFDL-ESM2G, HadGEM2-ES, IPSL-CM5A-LR, MIROC-ESM, MIROC-ESM-CHEM, MIROC5, MPI-ESM-LR, MPI-ESM-MR, MPI-CGCM3, NorESM1-M, and CSIRO-Mk3.6.0) with different runs, altogether 31 outputs, are used to project changes of high temperature for 1.5 °C and 2.0 °C global warming, relative to the reference period (Supplementary Table 2). The GCM outputs were bias-corrected and downscaled statically to a regular geographical grid of 0.5° resolutions, based on observations, to show the GCMs have a good consistency in simulating high temperature in the major cities of China (Supplementary Fig. 2).

**Population and GDP**. County-level population and GDP in China for 1986–2017 are from the Statistical Yearbook of China. Based on the most recent Sixth Population Census in 2010 and the latest universal two-child policy, the parameters of the Population-Development-Environment model are regionalized to project population under Shared Socioeconomic Pathways (SSPs) in China for the 21st century[50,51]. The GDP in China under SSPs is projected with regionalized parameters using the Cobb-Douglas production model[52,53], and is standardized to 2010 price level to maintain the homogeneity of data series. All the GDP and population are projected at the provincial scale first. Then, based on the county-level distribution of population and GDP in 2010, the area ratio method is applied to downscale population and GDP into the 0.5° resolution. Finally, the population and GDP within the boundaries of the city are summed.

**Distributed lag non-linear model**. The temperature-mortality relationship is set up using a distributed lag non-linear model, which can describe complex non-linear and lagged dependencies through the combination of the conventional exposure-response association and the additional lag-response association[54].

A natural cubic B-spline of time with 8 degrees of freedom per year is applied to control long-term trends and to indicate the days of a week[55]. The lag-response association represents the temporal change in risk after a specific exposure, and estimates the distribution and delayed effects that cumulate across the lag period. We modeled the exposure-response curve with a quadratic B-spline with three internal knots placed at the 10th, 75th, and 90th percentiles of location-specific temperature distributions, and the lag-response curve with a natural cubic B-spline with an intercept and three internal knots placed at equally spaced values in the log scale. We extended the lag period to 10 days to include the long delay of the high temperature effects as it usually lasts around a week[36,56–58]. The fitted meta-analytical model is used to derive the best linear unbiased prediction of the overall cumulative temperature and mortality association, and the minimum mortality temperature. We define the minimum mortality temperature as the threshold temperature.

$$log[E(Y_t)_s] = \alpha + \beta * Temp, l + NS(Time, df) + \gamma * Dow + \delta * Holiday \quad (1)$$

$$RR_{I,s} = exp(\beta * I_s)s = 1, 2, 3, \ldots \ldots, 27 \quad (2)$$

Where $E(Y_t)$ is the observed daily mortality at calendar day $t$; $l$ refers to the maximum lag days, and $Temp$, $l$ is the cross-basis matrix for the two dimensions of maximum temperature and lags; the natural cubic spline function $NS()$ captures the non-linear relationship between the covariate (time) and mortality; $Dow$ and $Holiday$ are the dummy variables for the day of the week and public holiday; $RR_{I,s}$ is the relative risk corresponding to high temperature with certain intensity for metropolises, and greater or equal to 1; $I$ is the intensity of high temperature, deduced by difference between daily maximum temperature and the minimum mortality temperature; and $s$ represents the different metropolises.

All analyses were performed using the R software Version 3.5.1 (R Foundation for Statistical Computing, Vienna, Austria) by using DLNM and MVMETA packages.

**Projection of future heat-related mortality**. Heat-related mortality at 1.5 °C and 2.0 °C global warming are projected by combining the simulated daily maximum temperature and the temperature-mortality relationship. We computed city-specific heat-related mortality as follows:

$$M_s = Y_s \times ERC_{I,s} \times POP_s \quad (3)$$

$$ERC_{I,s} = RR_{I,s} \times (1 - AC_I) - 1 \quad (4)$$

where $s$ represents the different metropolises, $I$ is the intensity of high temperature; $M_s$ is the daily heat-related mortality; $Y_s$ represents daily mortality rate per million in the observational period; $POP_s$ is the population; $ERC_{I,s}$ is the increase in relative risks along with intensification of high temperature, which is related to the improved adaptation capacity $AC_I$ (Supplementary Fig. 5).

## Data availability

The dataset generated and analyzed during this study are available (with some institutional limitations) from the corresponding authors upon reasonable request. The source data underlying Figs. 1a–b, 2, and 3a–b are provided as a Source Data file.

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

## Acknowledgements

This study was jointly supported by the National Key Research and Development Program of China MOST (2018FY100501) and the Cooperation Project between the Natural Science Foundation of China and the Pakistan Science Foundation (41661144027). The Climate Change Science Fund of the China Meteorological Administration (CCSF 201722, 201810, 201924) provides a policy-oriented training course for PhD students. This work and its contributors were supported by the UK-China Research & Innovation Partnership Fund through the Met Office Climate Science for Service Partnership (CSSP) China as part of the Newton Fund. The authors are thankful for the support by the High-level Talent Recruitment Program of the Nanjing University of Information Science and Technology (NUIST). The authors would like to thank the World Climate Research Program's working group on coupled modeling for producing and making available their model output.

## Author contributions

T. Jiang and Z.W. Kundzewicz conceived the study. Y.J. Wang, A.Q. Wang and J.Q. Zhai contributed equally to this paper by performing analyses and drafting the paper. T. Fischer and B.D. Su integrated innovative ideas and modified the complete research and manuscript. Q.Y. Liu and J. Yang investigated the mortality data for 27 metropolitans in China. M.J. Zhan and H. Tao downscaled and bias corrected the 31 GCMs maximum temperature data. G.J. Wang analyzed the high temperature for 27 metropolitans. C. Gao and Z.Q. Feng set up the regionalized SSPs. All authors discussed the results and edited the manuscript.
