## [Peer Review File · Nature Communications]

Reviewers' comments:

Reviewer #1 (Remarks to the Author):

Reviewer's comments

Overall Comments

This manuscript assessed the heat-related morbidity and mortality in 27 Chinese megacities under highly concerned global warming of 1.5 °C and 2.0 °C. The authors' attempt to address the policy-relevant global warming targets is of great interest to the global climate change community. However, there are several major methodological concerns that prevent this paper from being published in *Natural Communications*.

Major Comments

The major concern for this manuscript is the blurry description of the key information in estimating the climate change-induced heat-related health burdens and inappropriate usage of statistical methods.

1. Regarding the 1.5 °C and 2.0 °C global warming scenarios, it seems that RCP2.6 around 2030 was used to represent the 1.5 °C scenario and RCP4.5 around 2050 were used to represent the 2.0 °C scenario (lines 131-139). This may be a big problem as the policy-relevant 1.5 or 2.0 °C in the Paris Agreement refers to limiting the global warming of 1.5 or 2.0 °C by 2100, rather than reaching the limit in a certain year during the 21st century. Obviously, even the high emission scenario RCP8.5 in a certain future year reached the limit of 1.5°C and 2.0°C global warming. Thus this approach by using RCP2.6 around 2030 and RCP4.5 around 2050 for the 1.5 °C scenario and 2.0 °C scenario is inappropriate.

It is also very unclear exactly which year in the future was used in RCP2.6 and RCP4.5, which should be provided as basic information. If different future period was used for RCP2.6 (around 2030?) and RCP4.5 (around 2050?), the comparison between 1.5 °C scenario and 2.0 °C scenario is problematic and misleading. Thus, reanalysis or re-run the GCMs for the specific 1.5 °C and 2.0 °C scenario is highly recommended.

Furthermore, as the 1.5 °C and 2.0 °C are the global average temperature increases, it would be interesting to see the future temperature increases in China under those scenarios.

2. Another major concern is about the heat-related mortality-/morbidity-inducing events. It is not clear how these events were calculated. Previous epidemiological studies generally found that the

heat-related cardiovascular mortality and morbidity that were not originally identified in the death certificates or hospital records, were much larger than those originally identified (e.g., heat stroke). If only originally identified mortality and morbidity in records were used, it may largely underestimate the heat-related health burden. Besides, it is also not clear which cause of death was used for mortality and which type of morbidity data (e.g., hospital admissions, emergency hospital room visits) were used for morbidity.

3. The piecewise regression methods for estimating the temperature threshold of heat-related mortality and morbidity also raises concerns about the nonlinear effect of temperature on mortality and morbidity and missing adjustment for confounders. As most recent temperature-mortality/morbidity studies found nonlinear effects of temperature, using a linear assumption may significantly bias the estimates. In addition, important confounders such as temporal varying confounders and the day of week effect must be controlled in estimating the heat-related health effects. More standard time-series regressions such as distributed lag nonlinear models (DLNM) are suggested to estimate the heat-related health effects and heat thresholds.

4. Another confusing issue is the misuse of “heat-related” mortality/morbidity while this paper actually assessed the heat wave effects rather than heat (high temperature) effects. Heatwave effects are defined differently than heat effects. In addition, it is very confusing to tell the difference between the frequency, intensity, and duration of heat-related morbidity and mortality events and those characteristics of heat-wave events.

5. The “duration-mortality/morbidity rate function” used to project future heat-related health burdens is too simplistic. Longer duration (e.g., more heat wave days) certainly means a larger number of health burden compared with short duration. In addition to duration, previous studies also showed that intensity of heat waves and the occurring time in summer (i.e., earlier heat waves have higher mortality risks) also contribute to the health burdens of heat waves. The intensity and occurring time of heat waves should also be considered in projecting the health burden of future heat waves.

6. The source of five SSP population projections at 0.5° resolution is missing. It is not clear whether the authors just used the 0.5° resolution SSP population projections for the total population or the age group-specific and gender-specific SSP projections. To the best of my knowledge, the gender and age-specific SSP projections are only available now at country level while total population SSP projections are available at high spatial resolutions like 0.5°C. If only total population projections were applied, this would be a major limitation that should be discussed in the paper.

7. The results described in lines 185-232 should be revised to make it more clear and concise. Instead of a detailed description of each SSP, temperature scenario, health outcome, a table summarizing the results is highly recommended.

8. In the discussion, no comparisons between the estimated heat-related morbidity and mortality burdens in this study and in previous studies were conducted. The temperature threshold was also not discussed and compared with previous findings. These comparisons are important as it is not sure that the estimated heat-related burdens in this paper are correct and precise.

9. Previous studies showed that adaptation may play a major role in projecting heat-related mortality under climate change. The limitation of not considering human adaptation to heat effects should be discussed.

Minor Comments

1. The title should be revised to make it more clear that it assessed the health impacts of heat waves rather than heat (i.e., high temperatures).

2. Line 17. The unit of morbidity and mortality should be clarified.

3. Line 29 and the title. The conclusion of "...increase by about 50%" is obviously wrong. The percent of avoided health burden in the 1.5 °C scenario compared with the 2.0 °C scenario should be $(166-114)/166=31\%$ for morbidity and $(60-38)/60=36.7\%$ for mortality. If comparing with the 1.5 °C scenario, the 2.0 °C scenario will lead to additionally $(166-114)/114=45.6\%$ morbidity and $(60-38)/38=57.9\%$ mortality. Thus, heat-related morbidity and mortality don't double but increase by ~50% comparing the 2.0 °C with 1.5 °C. The title should be revised for the conclusion of 'double between 1.5°C and 2.0°C'.

4. Line 58-60. Citations are missing for this statement that warming in China will be faster than the global average continuously.

5. Line 266-268. This statement may be wrong as SSP population projections in China will decrease by the end of this century. This statement should be revised by adding the future time period used in this research.

6. Fig3. As mentioned earlier, it is confusing to tell if this is the figure for heat waves or heat-related mortality and morbidity events.

7. Line 320. The “Heat-related morbidity and mortality data” should be in boldface and a blank space is missing after this term.

8. Line 365-366. Citation missing for this statement.

Reviewer #2 (Remarks to the Author):

The authors presented heat-related morbidity and mortality in China’s megacities under the 1.5 and 2.0 degree scenarios. They concluded that 1.5 target can avoid the heat-related morbidity and mortality by about 50%. This manuscript provides useful information by up-to-date methodologies using RCP and SSP simultaneous. Most of the methodologies and results are effectively demonstrated. I have some comments and questions.

1. The authors should provide more information on morbidity and mortality data. To construct morbidity data is a challenging procedure in many countries. Some questions are: what kind of diseases were included? Providing ICD (international code of disease) would be helpful for readers. Did you include out-patients, emergency room visits as well as scheduled visits? Are there any references to explain these procedures? Chinese literatures are acceptable for this purpose. Similar problems for mortality data. Did you include all causes or exclude accidental deaths? Who is responsible for these data in each city?

2. I have concerns on the model validity. Time series data only for 3-year does not provide a reliable exposure-response relationship in general. I’d like to see the scatter plots between temperature and morbidity and temperature and mortality with smoothing lines to check the validity of the piece-wise linear model for all cities, especially for small cities. I think the estimated thresholds (31.5~37.7) are too high so that the frequency of heat-related morbidity and mortality inducing events are too small. As a result, estimated morbidity and mortality are also too small. Thresholds in Figure 4 of Hajat and Kosatky’s 2010 paper are between 16-33 degrees.

3. I assumed that 25,384 and 8,479 in line 17 are annual total. Is this right? I found similar problems in other places of the manuscript. Please put “annual” when appropriate.

4. I’m a little bit uncomfortable about future time (year). 25,384 and 8,479 in line 17 are projected values with 1.5 degree. How about year? Population sizes are very different for 2050 and 2100 under any SSPs. Population size of China is expected to increase to a certain year then drop rapidly after that. I think population structure and size are more dependent on the time rather than SSP.

5. The authors can add more information on Fig S1. My suggestions are: use colors indicating mean temperature of the city (red for hot city, blue for cold city), diameter proportional to the population size (big dot for big city)

6. The authors can add more information on Table S2. My suggestions are population size, annual morbidity and mortality (not heat-related but reported total), mean temperature.

Reviewer #3 (Remarks to the Author):

Summary: The authors present an analysis of the current and projected health burden of heat extremes in megacities of China. The topic is of obvious importance, and the authors present a large number of results that are supported by a significant dataset on heat-related health impacts. As such, both the topic and some of the specific results will be of interest to the research community. As written, however, the paper is not acceptable for publication in Nature Communications. Key methodological points require more explanation, results are presented in bursts of numbers that sometimes lack statistical significance tests and often lack sufficient context to isolate the “key” result out of the many numbers presented, and the flow of the text is difficult to follow. This final point might have something to do with English grammar, but I don’t want to unduly penalize the authors for language issues. I think that the larger issue is with logical flow—it’s often hard to anchor oneself within a section because it’s not clear what the text is trying to accomplish.

In this sense, my feeling that the text is hard to read is linked to my single largest concern about the manuscript: it’s still not clear to me exactly why the authors pursued the analyses that they did.

There are interesting questions in here related to 1.5C vs. 2.0C stabilization, to differential impacts by age group and sex, to geography, and (perhaps) to implications of different SSP. But in the jumble of results on all of these topics, some significant and some not significant, I lose sight of why the authors have decided to look at this particular set of factors. A clearer statement of purpose and testable hypotheses—and perhaps the removal of some ancillary results—would enhance the impact of the paper.

Major comments:

1. I don't understand the title. It refers to doubling of impacts between 1.5C and 2.0C stabilization targets, but I don't see an actual doubling in any of the major results presented in the paper.
2. What are we supposed to take away from the multiple SSP comparison? I could see how this might be a useful analysis, in that SSP present different visions of how climate stabilization is achieved, and the authors suggest that including SSP analysis is a key strength of their work. But I have trouble seeing how the SSP results are interesting in the end. Are there key differences that I'm reading past? Or is there something interesting to be concluded from the lack of major differences between SSP? The "Discussion and Conclusions" section simply restates the SSP results without offering any real explanation or interpretation. As far as I can tell the SSP angle could be completely dropped from this paper, since the results don't show any differences that the authors find worth explaining.
3. Definition of heat wave thresholds: In Supplementary Material the authors list the thresholds used for each city and indicate that they were arrived at through piecewise regression on J-shaped mortality curves. This piecewise regression analysis should be included in the Supplementary Material. Arriving at a single breakpoint in this kind of analysis can be very difficult, and the authors should either demonstrate how they were able to arrive at single specific values that differ by tenths of a degree between cities or they should present some kind of sensitivity analysis on the choice of break points.
4. Duration-Morbidity-Mortality curves: Do I understand correctly that a single temperature threshold was used to define whether an event was morbidity or mortality inducing, and that the rate of morbidity and mortality across sex and age group was then determined as a sex and age-group specific linear response to duration? If so, then can the authors justify why a single break point was used across all groups when it is clear that they are responding differently to heat? And

why is it that duration but not intensity is used to define the morbidity and mortality rates in each sex and age group?

5. The authors state that they downscaled and bias corrected GCM output to 0.5 degrees. For any analysis of climate extremes the choice of downscaling method can be very important. How was downscaling performed here? Were second and higher moment statistics of the GCM temperature distribution corrected to observation?

6. Introduction: I suggest a complete rewrite. Paragraph 2 doesn't say much and can be dropped. Paragraph 3 suggests that SSP and sex/age specificity will be the primary contribution of this study. Paragraph 4 talks about urbanization. Paragraph 5 talks about Paris targets. This is wide-ranging and lacks focus, and given the brevity of each paragraph the introduction really doesn't do justice to any of the topics. If I were writing this paper I would move Paris targets to Paragraph 2. One of the most compelling motivation for this study (to my read) is to understand the implications of 1.5 vs. 2.0 stabilization targets—there is a serious, policy-relevant result on this topic that the paper provides. I would then follow the over-arching Paris target motivation with the recognition that impacts will vary across groups and by geography, which motivates the sex/age/north-south aspects of the study. Personally, I don't think you even need to get into urban effects here, as nobody can argue that it's not important to look at health in megacities of China. Obviously, this is just one suggestion. My main concern is that the introduction should clearly motivate the top analyses performed in the paper. Right now it's more like a listing of important climate issues, and much of the results section that follows is similarly broad and hard to catch hold of.

7. Results: I acknowledge that style of results presentations varies across fields, and that health literature leans towards large numbers of confidence intervals presented in text and extensive tables where simple-minded climate scientists like me want to see results boiled down into graphs and summary tables. But since this paper is being submitted to Nature Communications as a high-impact climate and health paper, I do think that some structure and distillation of results is necessary. I would ask the authors to remove lists of results that do not yield interesting differences and that are not interpreted in a meaningful way in the Discussion section. Instead, focus on the results that do yield insights, make sure that they are supported with appropriate statistical tests, and, where possible, highlight them in a figure or table on key findings. I personally get completely bogged down in these long lists of results, many of which appear to be statistically insignificant.

Additional specific comments:

1. I hate to harp on grammar, but the manuscript is difficult to follow in some passages. The authors or the journal editorial experts should attempt to edit grammar for clarity. As one example,

the authors repeatedly employ the phrase “might be” when talking about projections. I’m not sure what is implied by that choice of words. I would opt for more precise language, such as “is projected to be between,” or some similar phrasing. There are other examples like this throughout the text.

2. Line 110: This aside about definition is distracting. Either put it at the beginning of the presentation of results or let the reader find it in the methods section.

3. Line 117: Why aren’t the 2006-2015 mortality results presented? The authors list the figures for morbidity in the previous paragraph but do not list the corresponding numbers for mortality here.

4. Figure 2: This figure seems unnecessary. The four numbers that go into the large bar chart are already stated in the text. It’s also not clear why the results from 2011-2013 are plotted, when much longer periods are discussed in the text. More broadly, I think this figure could be usefully replaced by a more informative table that summarizes the many 1986-2005 vs. 2006-2015 numbers on heat wave duration, intensity, and frequency that are listed in the text. A table would help the reader keep these numbers organized in his/her mind.

5. Figure 3: This figure should be supported by statistical tests of the trends the authors describe in the text. Are any of the trends significant?

6. Figure 4: The authors should provide appropriate statistical significance tests on the changes summarized in this figure and listed in the text throughout the section. Any time that North and South are contrasted in a quantitative way or a claim is made about present day vs. 1.5C warming vs. 2.0C warming a statistical test should really be provided.

7. Line 168: To the reader unfamiliar with SSP, this choice is entirely mysterious. How did the authors decide which SSP to associate with which RCP? I would suggest a paragraph or two on the SSP in the methods or supplementary materials, as I think that many readers interested in this study will not be familiar with the SSP.

Responses to the reviewers' comments

Reviewer #1 (Remarks to the Author):

This manuscript assessed the heat-related morbidity and mortality in 27 Chinese megacities under highly concerned global warming of 1.5 °C and 2.0 °C. The authors' attempt to address the policy-relevant global warming targets is of great interest to the global climate change community. However, there are several major methodological concerns that prevent this paper from being published in *Natural Communications*.

The major concern for this manuscript is the blurry description of the key information in estimating the climate change-induced heat-related health burdens and inappropriate usage of statistical methods.

Major Comments

1. Regarding the 1.5 °C and 2.0 °C global warming scenarios, it seems that RCP2.6 around 2030 was used to represent the 1.5 °C scenario and RCP4.5 around 2050 were used to represent the 2.0 °C scenario (lines 131-139). This may be a big problem as the policy-relevant 1.5 or 2.0 °C in the Paris Agreement refers to limiting the global warming of 1.5 or 2.0 °C by 2100, rather than reaching the limit in a certain year during the 21st century. Obviously, even the high emission scenario RCP8.5 in a certain future year reached the limit of 1.5°C and 2.0°C global warming. Thus this approach by using RCP2.6 around 2030 and RCP4.5 around 2050 for the 1.5 °C scenario and 2.0 °C scenario is inappropriate.

It is also very unclear exactly which year in the future was used in RCP2.6 and RCP4.5, which should be provided as basic information. If different future period was used for RCP2.6 (around 2030?) and RCP4.5 (around 2050?), the comparison between 1.5 °C scenario and 2.0 °C scenario is problematic and misleading. Thus, reanalysis or re-run the GCMs for the specific 1.5 °C and 2.0 °C scenario is highly recommended.

Furthermore, as the 1.5 °C and 2.0 °C are the global average temperature increases, it would be interesting to see the future temperature increases in China under those scenarios.

Answer:

The global mean temperature in 1986-2005 was 0.61°C warmer than that in the pre-industrial period (1850-1900), and a further warming of 0.89°C and 1.39°C indicate the 1.5°C and 2.0°C limits above the pre-industrial level, respectively.

An ensemble mean of multiple GCMs of the Coupled Model Intercomparison Project phase 5 (CMIP5) shows that a 20-year moving average of global mean temperature (GMT) will reach the 1.5°C warming threshold after 2030 under RCP2.6, and reach the 2.0°C after 2050 under RCP4.5. Projected temperature will be in low variation since 2060s under both scenarios (Warszawski et al., 2014; Su et al., 2017; Sun et al., 2017). In order to conduct an impact study in a comparative stable climatic condition, we choose the RCP2.6 scenario for 2060-2099 as the 1.5°C warming world, but RCP4.5 scenario for the same period as the 2.0°C warming, although there will be overshoot (P. 3, L. 88-98).

2. *Another major concern is about the heat-related mortality-/morbidity-inducing events. It is not clear how these events were calculated. Previous epidemiological studies generally found that the heat-related cardiovascular mortality and morbidity that were not originally identified in the death certificates or hospital records, were much larger than those originally identified (e.g., heat stroke). If only originally identified mortality and morbidity in records were used, it may largely underestimate the heat-related health burden. Besides, it is also not clear which cause of death was used for mortality and which type of morbidity data (e.g., hospital admissions, emergency hospital room visits) were used for morbidity.*

Answer:

The daily mortality data during 2007-2013 were collected from the Chinese National Center for Chronic and Non-communicable Disease Control and Prevention. The underlying cause of death was coded based on the 10th Revision of the International Statistical Classification of Diseases and Related Health Problems (ICD-10). Daily non-accidental mortality (ICD-10: A00-R99), mortality due to cardiovascular disease (I00-I99), respiratory disease (J00-J99), and so on, were further categorized into four groups by age and gender: working age (age: 15-64) and non-working age (age: ≥ 65 and ≤ 14); female and male. Details of the mortality data can be found in the previous study by Yang et al. (2019).

Morbidity data in this study are based on hospitalization records. Annual number of hospitalizations in each province was recorded in the China Statistical Yearbook. The number of hospital admission in each city is deduced based on the proportion of the city population to the provincial total. We allocate the annual hospital admission into the daily scale, by assuming that the percentage of mortality to morbidity remains unchanged within a year (P. 9, L. 296-300).

3. *The piecewise regression methods for estimating the temperature threshold of heat-related mortality and morbidity also raises concerns about the nonlinear effect of temperature on mortality and morbidity and missing adjustment for confounders. As most recent temperature-mortality/morbidity studies found nonlinear effects of temperature, using a linear assumption may significantly bias the estimates. In addition, important confounders such as temporal varying confounders and the day of week effect must be controlled in estimating the heat-related health effects. More standard time-series regressions such as distributed lag nonlinear models (DLNM) are suggested to estimate the heat-related health effects and heat thresholds.*

Answer:

It is true that the piecewise linear regression method (which, in fact, is a very simple nonlinear method, yet being piecewise linear) may not adequately represent the nonlinear effect of temperature on morbidity and mortality, and a piecewise linear assumption may significantly bias the estimates. In the revised manuscript, we considered morbidity and mortality data for the period 2007-2013 (originally from 2011-2013 only). The Distributed Lag Non-linear Model (DLNM), proposed in literature, was applied in the revised draft to

estimate the health effects of high temperature (P. 9-10, L. 317-345). Examples from six cities are plotted to show the relationship between the relative risk of morbidity and mortality and the daily maximum temperature (Supplementary Fig. 3 and 4).

4. Another confusing issue is the misuse of “heat-related” mortality/morbidity while this paper actually assessed the heat wave effects rather than heat (high temperature) effects. Heat wave effects are defined differently than heat effects. In addition, it is very confusing to tell the difference between the frequency, intensity, and duration of heat-related morbidity and mortality events and those characteristics of heat-wave events.

Answer:

The purpose of this study is to assess the changes in heat-related morbidity and mortality under a 1.5°C and 2.0°C global warming. The "heat-related" morbidity/mortality in the original manuscript might be misleading. In the revised manuscript, we adopt the DLNM model to assess the relative risk of morbidity/mortality for any given temperature, and the lowest morbidity-inducing temperature is defined as the threshold temperature. Once daily maximum temperature reaches and exceeds the threshold, the days with high-temperature start to be counted. The intensity of high temperature is defined as the amplitude of temperature above the threshold. The frequency and intensity of high temperature under global warming scenarios are introduced in the main text (P. 3-4, L. 113-119). The terms: frequency, intensity, and duration of heat-related morbidity and mortality events are deleted.

5. The “duration-mortality/morbidity rate function” used to project future heat-related health burdens is too simplistic. Longer duration (e.g., more heat wave days) certainly means a larger number of health burden compared with short duration. In addition to duration, previous studies also showed that intensity of heat waves and the occurring time in summer (i.e., earlier heat waves have higher mortality risks) also contribute to the health burdens of heat waves. The intensity and occurring time of heat waves should also be considered in projecting the health burden of future heat waves.

Answer:

In the revised manuscript, the Distributed Lag Non-linear Model (DLNM) is set up for each major city of China, based on the morbidity and mortality cases and observed temperature during 2007-2013, and the relative risk corresponding to high temperature can be deduced. The DLNM can simultaneously estimate the non-linear effects of temperature at each lag and the non-linear effects across these lags. The lag period is extended to 10 days in this study to include the long delay of the high-temperature effects (see data and method section). We assume a stationarity of the temperature-morbidity / mortality relationship. Indeed, we are aware that the time of occurrence of a heat wave may play a role. Early heat waves may induce higher morbidity and mortality because of lack of acclimatization. However, we have not studied this track and did not consider this effect in our paper.

6. The source of five SSP population projections at 0.5 ° resolution is missing. It is not

clear whether the authors just used the 0.5° resolution SSP population projections for the total population or the age group-specific and gender-specific SSP projections. To the best of my knowledge, the gender and age-specific SSP projections are only available now at country level while total population SSP projections are available at high spatial resolutions like 0.5°C. If only total population projections were applied, this would be a major limitation that should be discussed in the paper.

Answer:

We predict the population in the 21st century for all SSPs using the Population-Development- Environment model, based on the Sixth Population Census in 2010 and the latest developments in the two-child policy in China. For SSP1 it was assumed that population will be in low fertility, low mortality, medium migration, and high education; for SSP2 – medium fertility, medium mortality, medium migration and medium education; for SSP3 – high fertility, high mortality, low migration and low education; for SSP4 – high fertility, high mortality, medium migration, and polarized education; and for SSP5 – low fertility, low mortality, high migration, and high education. The annual GDP of China under all SSPs is updated by applying the Cobb-Douglas production model with regionalized parameters, a newly predicted labor force, and with standardized prices to 2015 to maintain the homogeneity of the data series.

At first, population and GDP are both predicted at the provincial scale. Then, the area ratio method is applied to downscale the predicted population and GDP into 0.5° resolution based on the spatial distribution of county-level population and GDP in 2010. Finally, the population and GDP within the boundaries of a metropolis are summed-up (Supplementary Information SI-5).

7. The results described in lines 185-232 should be revised to make it more clear and concise. Instead of a detailed description of each SSP, temperature scenario, health outcome, a table summarizing the results is highly recommended.

Answer:

The combination of SSPs with climate scenarios produces a range of heat-related mobility/mortality risks. SSPs describe settings of plausible alternative futures of societal development, which consider the effects of climate change and climate policies over the 21st century. The description has been greatly simplified by reducing the main text and adding two tables in the Supplementary (Supplementary Table 4 and 5), as per recommendation of this reviewer.

8. In the discussion, no comparisons between the estimated heat-related morbidity and mortality burdens in this study and in previous studies were conducted. The temperature threshold was also not discussed and compared with previous findings. These comparisons are important as it is not sure that the estimated heat-related burdens in this paper are correct and precise.

Answer:

In the revised version, comparing our results with previous studies now enriches the discussion section. For example we added: increases in heat-related annual mortality in Jiangsu province in eastern China was estimated at 102 per million for 2041-2065 under RCP4.5 relative to 1981-2005 (Chen et al., 2016). Similarly, Petkova et al. (2014) estimated heat-related annual mortality of 134 per million in New York and 107 per million in Philadelphia under RCP4.5 for 2070-2099 relative to 1971-2000. Our estimation results on mortality in China metropolises in the last forty years of the 21st century are higher than those estimated for the US, which indicates a lower adaptation capacity in China than in the US. Of course, other factors, such as the differences in climate models, emission scenarios as well as baseline mortality rates, also contribute to the differences in mortality estimation (P. 7, L.222-242).

9. Previous studies showed that adaptation may play a major role in projecting heat-related mortality under climate change. The limitation of not considering human adaptation to heat effects should be discussed.

Answer:

With the socio-economic development, the adaptive capacity of human beings to climate change is bound to improve. In the revised manuscript, GDP per capita is used as an indicator of the socio-economic development to assess changes of adaptation capacity (Supplementary Fig. 6 and Fig. 7).

Minor Comments

1. The title should be revised to make it more clear that it assessed the health impacts of heat waves rather than heat (i.e., high temperatures).

Answer:

The title of the manuscript is changed to “Tens of thousands of additional deaths in major cities of China between the 1.5 °C and 2.0 °C warming”

2. Line 17. The unit of morbidity and mortality should be clarified.

Answer:

To allow a rough comparison with previous studies, we computed the heat-related morbidity and mortality as the number of sick people and fatalities per million people (unit: number of people per million of people), throughout the paper.

3. Line 29 and the title. The conclusion of “...increase by about 50%” is obviously wrong. The percent of avoided health burden in the 1.5 °C scenario compared with the 2.0 °C scenario should be $(166-114)/166=31\%$ for morbidity and $(60-38)/60=36.7\%$ for mortality. If comparing with the 1.5 °C scenario, the 2.0 °C scenario will lead to additionally $(166-114)/114=45.6\%$ morbidity and $(60-38)/38=57.9\%$ mortality. Thus, heat-related morbidity and mortality don't double but increase by ~50% comparing the 2.0 °C with 1.5 °C. The title should be revised for the conclusion of ‘double between 1.5°C and 2.0°C’.

Answer:

The title and the conclusion have been revised according to the suggestions. The Conclusion chapter has been changed in lines 21-29.

4. Line 58-60. Citations are missing for this statement that warming in China will be faster than the global average continuously.

Answer:

A reference is added in the main text to clarify the source of the statement that warming in China will be faster than the global average (P.2-3, L71-72).

5. Line 266-268. This statement may be wrong as SSP population projections in China will decrease by the end of this century. This statement should be revised by adding the future time period used in this research.

Answer:

Population of 27 major cities is predicted under SSPs for the entire 21st century. In order to match the socio-economic scenarios with climate scenarios, the prediction of the population for 2060-2099 is used in the impact study of 1.5°C and 2.0°C global warming in our paper. Detailed information can be found in Supplementary Information SI-5.

6. Fig3. As mentioned earlier, it is confusing to tell if this is the figure for heat waves or heat-related mortality and morbidity events.

Answer:

In the revised manuscript, we focus on analyses of the heat-related morbidity and mortality at 1.5°C and 2.0°C global warming. The changes in frequency and intensity of high temperature are briefly introduced in the main text (P. 3-4, L. 113-125).

7. Line 320. The “Heat-related morbidity and mortality data” should be in boldface and a blank space is missing after this term.

Answer:

The format and spelling mistakes have been checked and revised throughout the manuscript.

Reviewer #2 (Remarks to the Author):

The authors presented heat-related morbidity and mortality in China's metropolitans under the 1.5 and 2.0 degree scenarios. They concluded that 1.5 target can avoid the heat-related morbidity and mortality by about 50%. This manuscript provides useful information by up-to-date methodologies using RCP and SSP simultaneous. Most of the methodologies and results are effectively demonstrated. I have some comments and questions.

Comments

1. The authors should provide more information on morbidity and mortality data. To construct morbidity data is a challenging procedures in many countries. Some questions are: what kind of diseases were included? Providing ICD (international code of disease) would be helpful for readers. Did you include out-patients, emergency room visits as well as scheduled visit? Are there any reference to explain these procedures? Chinese literatures are acceptable for this purpose. Similar problems for mortality data. Did you include all causes or exclude accidental deaths? Who is responsible for these data in each city?

Answer:

Detailed information on morbidity and mortality data is now described in the data and method section. We used daily non-accidental mortality (ICD-10: A00-R99), mortality due to cardiovascular disease (I00-I99), respiratory disease (J00-J99), and so on. These were further categorized these into four groups by age and gender: working age (age: 15-64) and non-working age (age: ≥ 65 and ≤ 14); female and male. The details of the mortality data can be found in the previous study by Yang et al. (2019).

The daily mortality data in China metropolises during 2007-2013 were collected from the Chinese National Center for Chronic and Non-communicable Disease Control and Prevention. The underlying cause of death was coded based on the 10th Revision of the International Statistical Classification of Diseases and Related Health Problems (ICD-10). Amongst, daily non-accidental mortality (ICD-10: A00-R99), mortality due to cardiovascular disease (I00-I99), respiratory disease (J00-J99), and so on were further categorized into four groups by age and gender: working age (age: 15-64) and non-working age (age: ≥ 65 & ≤ 14); female and male. Details of the mortality data can be found in a previous study by Yang et al. (2019).

Morbidity data in this study are based on hospitalization records. Annual number of hospitalizations in each province was recorded in the China Statistical Yearbook. The number of hospital admission in each city is deduced based on the proportion of the city population to the provincial total. We allocate the annual hospital admission into the daily scale, by assuming that the percentage of mortality to morbidity remains unchanged within a year. (P. 9, L. 296-300).

2. I have concerns on the model validity. Time series data only for 3- year does not provide reliable exposure-response relationship in general. I'd like to see the scatter plots between temperature and morbidity and temperature and mortality with smoothing lines to check the validity of piece-wise linear model for all cities, especially for small cities. I think the estimated thresholds (31.5~37.7) are too high so that frequency of heat-related morbidity and mortality inducing events are too small. As a result, estimated morbidity and mortality

are also too small. Thresholds in Figure 4 of Hajat and Kosatky's 2010 paper are between 16-33 degree.

Answer:

We extended the morbidity and mortality data from 2011-2013 to 2007-2013. Taking the non-linear effect of temperature on mortality/morbidity into consideration, the Distributed Lag Nonlinear Model (DLNM) is now applied to estimate the health effects of high temperature. The DLNM has recently been applied in studies to quantify the effects of temperature on human health. The major advantage of this model is that it simultaneously describes a non-linear exposure-response association and lagged effects. Supplementary Figs 3-4 show the relation between the relative risk of morbidity/mortality and the daily maximum temperature, which is set up by the DLNM method for 6 cities as example. Our threshold temperature is higher than findings in existing studies due to the fact that our estimation is based on daily maximum rather than daily mean temperature as many previous studies did (P. 7, L. 207-221). By the way, in our paper, we do not deal with smaller cities.

3. I assumed that 25,384 and 8,479 in line 17 are annual total. Is this right? I found similar problems in other places of the manuscript. Please put "annual" when appropriate.

Answer:

All amounts of morbidity and mortality are shown in annual scale at 1.5 and 2.0 °C global warming. The descriptions have been modified throughout the manuscript.

4. I'm a little bit uncomfortable about future time (year). 25,384 and 8,479 in line 17 are projected values with 1.5 degree. How about year? Population sizes are very different for 2050 and 2100 under any SSPs. Population size of China is expected to increase to a certain year then drop rapidly after that. I think population structure and size are more dependent on the time rather than SSP.

Answer:

According to the prediction of the population development in China under all SSPs, the population size will increase until 2030 and then decline during the rest of the 21st century. There are significant differences in the age structure and size of the population among the five SSPs. The socio-economic development of different SSPs also differs, which leads to differences in the improvement of the adaptation capacity to climate change. In our study, the results are presented for a range including all five SSPs. We refer to the common future time horizon 2060-2099 (RCP2.6 scenario for as the 1.5°C warming and RCP4.5 scenario as the 2.0°C warming, although there will be temperature overshoot).

5. The authors can add more information on Fig S1. My suggestions are: use colors indicating mean temperature of the city (red for hot city, blue for cold city), diameter proportional to the population size (big dot for big city)

Answer:

Fig. S1 in the original manuscript is re-plotted. Information on annual temperature of 27 metropolises during 1961-2015, and population in 2010 are added in Supplementary Fig. 1.

6. The authors can add more information on Table S2. My suggestions are population size, annual morbidity and mortality (not heat-related but reported total), mean temperature.

Answer:

We added Table 1 as attached below in the Supplementary Information according to the suggestions.

Table 1 Multi-year averaged summer temperatures for the period 1961-2015, the population in 2010, and the total annual average case numbers in morbidity and mortality for 27 major cities in China

Name	Averaged summer temperature (°C)	Population size (million)	Cases	
			total annual morbidity	total annual mortality
Beijing	30.5	13	1214366.00	69936.00
Changsha	32.3	3.3	433221.60	10221.86
Chengdu	28.6	6.2	874972.10	79874.00
Chongqing	30.7	19.5	2527327.00	168761.10
Fuzhou	31.4	2.2	235078.70	19613.43
Guangzhou	32.4	7.2	715340.90	41703.86
Guiyang	27.5	2.5	340658.60	26587.43
Harbin	26.3	1.8	482901.00	42634.14
Haikou	32.8	5	138763.90	7193.00
Hangzhou	28.5	4.2	477740.70	24112.14
Hefei	31.3	2.7	259629.10	8433.86
Hohhot	27.5	1.5	119104.40	11672.57
Jinan	31.4	3.9	458605.90	29919.29
Lanzhou	26.6	2.5	257790.10	9798.00
Nanchang	32.2	2.5	297030.90	7616.29
Nanjing	31.1	6.7	718619.10	22919.86
Nanning	32.6	3.1	383150.90	25371.29
Ningbo	31.3	2.5	241303.90	20998.29
Shanghai	30.4	14.3	1429944.00	56834.43
Shenyang	28.2	5.6	607717.70	33085.43
Shijiazhuang	31.5	2.8	273158.40	44192.86
Tianjin	29.8	8.7	654989.00	47506.43
Wuhan	31.9	1.2	710474.10	9022.86
Urumqi	30.1	4.9	453914.40	13185.29
Xi'an	31.7	6.2	711176.10	22006.86
Yinchuan	28.7	2.5	118708.10	5621.86
Zhengzhou	31.6	6.2	637222.00	13080.14

Reviewer #3:

The authors present an analysis of the current and projected health burden of heat extremes in megacities of China. The topic is of obvious importance, and the authors present a large number of results that are supported by a significant dataset on heat-related health impacts. As such, both the topic and some of the specific results will be of interest to the research community. As written, however, the paper is not acceptable for publication in Nature Communications. Key methodological points require more explanation, results are presented in bursts of numbers that sometimes lack statistical significance tests and often lack sufficient context to isolate the “key” result out of the many numbers presented, and the flow of the text is difficult to follow. This final point might have something to do with English grammar, but I don’t want to unduly penalize the authors for language issues. I think that the larger issue is with logical flow—it’s often hard to anchor oneself within a section because it’s not clear what the text is trying to accomplish.

In this sense, my feeling that the text is hard to read is linked to my single largest concern about the manuscript: it’s still not clear to me exactly why the authors pursued the analyses that they did. There are interesting questions in here related to 1.5C vs. 2.0C stabilization, to differential impacts by age group and sex, to geography, and (perhaps) to implications of different SSP. But in the jumble of results on all of these topics, some significant and some not significant, I lose sight of why the authors have decided to look at this particular set of factors. A clearer statement of purpose and testable hypotheses—and perhaps the removal of some ancillary results—would enhance the impact of the paper.

Answer:

Thank you for these constructive comments. We tried to react, *bona fide*, to all of them.

Major Comments

1. I don’t understand the title. It refers to doubling of impacts between 1.5C and 2.0C stabilization targets, but I don’t see an actual doubling in any of the major results presented in the paper.

Answer:

The title of the manuscript is now changed to include absolute numbers “Tens of thousands of additional deaths in major cities of China between the 1.5 °C and 2.0 °C warming”.

2. What are we supposed to take away from the multiple SSP comparison? I could see how this might be a useful analysis, in that SSP present different visions of how climate stabilization is achieved, and the authors suggest that including SSP analysis is a key strength of their work. But I have trouble seeing how the SSP results are interesting in the end. Are there key differences that I’m reading past? Or is there something interesting to be concluded from the lack of major differences between SSP? The “Discussion and Conclusions” section simply restates the SSP results without offering any real explanation or interpretation. As far as I can tell the SSP angle could be completely dropped from this paper, since the results don’t

show any differences that the authors find worth explaining.

Answer:

The SSPs describe a set of plausible alternative futures of societal development over the 21st century. There are significant differences in the age structure and size of the population among the five SSPs. Different economic development levels lead to differences in the improvement of adaptation capacity to climate change. In the revised version, all results are given in a form of a range of the five SSPs. We believe that the usage of SSPs is an essential asset of this paper. Using SSPs in estimating future morbidity and mortality is a big improvement in comparison to simplistic assumptions done in many other papers.

3. Definition of heat wave thresholds: In Supplementary Material the authors list the thresholds used for each city and indicate that they were arrived at through piecewise regression on J-shaped mortality curves. This piecewise regression analysis should be included in the Supplementary Material. Arriving at a single breakpoint in this kind of analysis can be very difficult, and the authors should either demonstrate how they were able to arrive at single specific values that differ by tenths of a degree between cities or they should present some kind of sensitivity analysis on the choice of break points.

Answer:

In the original manuscript, the piecewise linear regression method was used to calculate the threshold temperature. However, most recent temperature-mortality/morbidity studies proposed a more advanced way to represent nonlinear effects of temperature, and thus we now feel that using a piecewise linear assumption may significantly bias the estimations. In the revised version, the Distributed Lag Nonlinear Model (DLNM) was applied to define the morbidity and mortality inducing threshold temperatures. The DLNM can describe complex non-linear and lagged dependencies through the combination of the conventional exposure-response association and the additional lag-response association (Gasparrini and Leone, 2014).

As described in P. 9-10, L. 317-355: A natural cubic B-spline of time with 8 degrees of freedom per year is applied to control long-term trends and indicates the day of a week. Specific tutorials explain the technical details and terminology (Bhaskaran et al., 2013). The lag-response association represents the temporal change in risk after a specific exposure, and estimates the distribution and delayed effects that cumulate across the lag period. We modeled the exposure-response curve with a quadratic B-spline with three internal knots placed at the 10th, 75th, and 90th percentiles of location-specific temperature distributions, and the lag-response curve with a natural cubic B-spline with an intercept and three internal knots placed at equally spaced values in the log scale. We extended the lag period to 10 days to include the long delay of the effects of high-temperature as heat effect usually lasted around a week (Anderson and Bell, 2009; Wu et al., 2013; Gasparrini and Armstrong, 2013; Chen et al., 2016). The fitted meta-analytical model is used to derive the best linear unbiased prediction of the overall cumulative temperature and morbidity/mortality association, and the minimum morbidity/mortality temperature. We define the minimum morbidity/mortality temperature as the threshold temperature.

$$\log[E(Y_t)_s] = \alpha + \beta * Temp, l + NS(Time, df) + \gamma * Dow + \delta * Holiday \quad (1)$$

$$RR_{I,s} = \exp(\beta * I_s) \quad s = 1, 2, 3, \dots, 27 \quad (2)$$

where $E(Y_t)$ is the observed daily morbidity at calendar day t ; l refers to the maximum lag days, and $Temp, l$ is the cross-basis matrix for the two dimensions of maximum temperature and lags; the natural cubic spline function $NS()$ captures the non-linear relationship between the covariate (time) and mortality; Dow and $Holiday$ are the dummy variables for the day of the week and public holiday; $RR_{I,s}$ is the relative risk corresponding to high-temperature with certain intensity for metropolises, and greater than or equal to 1; I is the intensity of high-temperature, deduced by difference between the daily maximum temperature and minimum (threshold) morbidity/mortality temperature; s represents the different metropolises.

All analyses were performed using R software Version 3.5.1 (R Foundation for Statistical Computing, Vienna, Austria) by using DLNM and MVMETA packages.

Heat-related morbidity and mortality in the 1.5°C and 2.0°C global warming levels are projected by combining the daily maximum temperature in future and the temperature-morbidity/mortality relationship. We computed city-specific heat-related morbidity and mortality as follows:

$$M_s = Y_s \times ERC_{I,s} \times POP_s \quad (3)$$

$$ERC_{I,s} = RR_{I,s} \times (1 - AC_s) - 1 \quad (4)$$

where s represents the different metropolises; M_s is the daily heat-related morbidity/mortality; Y_s represents daily morbidity/mortality in the observational period; POP_s is the population; $ERC_{I,s}$ is the increase of relative risks along with intensification of high-temperature, which is related with the improved adaptation capacity AC_s (Supplementary Figs 6-7).

- Anderson B.G. and Bell M.L.,(2009) Weather-related mortality: how heat, cold, and heat waves affect mortality in the United States, *Epidemiology* **20**(2), 205-213.
- Bhaskaran K., Gasparrini A., Hajat S., Smeeth L. and Armstrong B.,(2013) Time series regression studies in environmental epidemiology, *Int J Epidemiol* **42**(4), 1187-1195.
- Chen K., Zhou L., Chen X. *et al.*,(2016) Urbanization Level and Vulnerability to Heat-Related Mortality in Jiangsu Province, China, *Environ Health Perspect* **124**(12), 1863-1869.
- Gasparrini A.,(2014) Modeling exposure-lag-response associations with distributed lag non-linear models, *Stat Med* **33**(5), 881-899.
- Gasparrini A. and Armstrong B., Reducing and meta-analysing estimates from distributed lag non-linear models. *BMC medical research methodology* (2013), p. 1.
- Gasparrini A., Guo Y., Hashizume M. *et al.*,(2015) Mortality risk attributable to high and low ambient temperature: a multicountry observational study, *The Lancet* **386**(9991), 369-375.
- Wu W., Xiao Y., Li G. *et al.*,(2013) Temperature–mortality relationship in four subtropical Chinese cities: A time-series study using a distributed lag non-linear model, *Science of The Total Environment* **449**, 355-362.

4. *Duration-Morbidity-Mortality curves: Do I understand correctly that a single temperature threshold was used to define whether an event was morbidity or mortality inducing, and that the rate of morbidity and mortality across sex and age group was then determined as a sex and age-group specific linear response to duration? If so, then can the authors justify why a single break point was used across all groups when it is clear that they are responding differently to heat? And why is it that duration but not intensity is used to define the morbidity and mortality rates in each sex and age group?*

Answer:

The Distributed Lag Nonlinear Model is applied now to deduce the threshold temperature of morbidity/mortality for each gender and age group (Supplementary Table 3). Accordingly, gender and age specific morbidity or mortality in [number per million people] is assessed for different metropolises under changing climatic and socioeconomic scenarios.

Table 3 Threshold temperate of heat-related morbidity and mortality for different gender and age groups in each metropolis

Name	Latitude (°N)	Threshold temperature (°C)							
		male		male		female		female	
		working		non-working		working		non-working	
		Mortality	Morbidity	Mortality	Morbidity	Mortality	Morbidity	Mortality	Morbidity
Beijing	39.9	28.4	27.6	27.4	26.5	27.4	26.5	26.8	25.9
Changsha	28.2	35.5	34.8	33.4	32.6	33.2	32.3	32.3	31.4
Chengdu	30.7	24.8	24.1	24.6	24.0	24.4	23.8	21.1	20.3
Chongqing	29.6	31.9	31.0	28.9	28.2	31.2	30.4	28.9	28.2
Fuzhou	26.1	33.3	32.7	32.8	32.1	33.1	32.3	32.7	31.9
Guangzhou	23.2	34.1	33.7	32.0	31.5	33.8	33.4	31.7	31.2
Guiyang	26.6	27.1	26.5	26.1	25.3	26.7	26.1	25.6	25.0
Harbin	45.8	29.1	28.1	27.5	26.5	28.1	27.2	26.9	26.0
Haikou	20	34.9	34.5	34.4	34.1	34.8	34.4	33.5	33.2
Hangzhou	30.3	32.2	31.3	30.8	29.9	31.3	30.4	30.1	29.2
Hefei	31.9	33.6	32.6	30.8	30.0	31.6	30.8	30.6	29.8
Hohhot	40.8	28.2	27.3	25.9	25.2	25.2	24.4	24.7	23.7
Jinan	36.7	31.9	31.1	30.1	29.6	31.1	30.4	29.6	28.8
Lanzhou	36	27.0	26.0	24.0	23.3	26.0	25.1	23.3	22.6
Nanchang	28.7	34.8	34.2	32.5	31.7	34.4	33.5	32.3	31.4
Nanjing	32	30.6	30.0	29.4	28.6	30.0	29.1	28.6	27.8
Nanning	22.8	33.9	33.4	33.0	32.6	33.8	33.3	32.5	32.1
Ningbo	29.9	31.6	30.7	30.1	29.3	30.7	29.8	29.6	28.7
Shanghai	31.2	31.4	30.6	30.0	29.2	30.6	29.8	29.5	28.7
Shenyang	41.8	28.6	28.0	28.2	27.5	28.4	27.8	27.8	27.1
Shijiazhuang	38	29.9	29.1	28.8	28.1	29.3	28.6	28.8	28.1
Tianjin	39.1	30.3	29.8	29.1	28.5	29.8	29.1	28.7	28.0
Wuhan	43.8	34.8	34.0	33.3	32.4	34.8	34.0	32.1	31.3
Urumqi	30.5	28.0	26.9	24.7	23.9	25.0	24.1	24.1	23.4

Xi'an	34.3	32.0	30.3	28.3	27.5	29.8	28.6	28.0	27.2
Yinchuan	38.5	30.9	29.9	30.6	29.7	30.9	29.9	30.6	29.7
Zhengzhou	34.8	33.8	32.8	32.5	31.8	33.1	32.3	31.8	31.1

5. The authors state that they downscaled and bias corrected GCM output to 0.5 degrees. For any analysis of climate extremes the choice of downscaling method can be very important. How was downscaling performed here? Were second and higher moment statistics of the GCM temperature distribution corrected to observation?

Answer:

Regarding the downscaling of GCMs and their simulation results, see details in supplementary information SI-2: Because there exists a bias between modeled and observed daily maximum temperature at each percentile, a method of Equidistant Cumulative Distribution Functions (EDCDF) is used to adjust the Cumulative Distribution Function (CDF) of simulation datasets. A normal distribution is fitted to the temperature field. The EDCDF method assumes that the difference between the observed and modeled values during the training period remains in the validation period for a given percentile. The EDCDF method can be written as:

$$\Delta = F_{oc}^{-1}(F_{ms}(x)) - F_{mc}^{-1}(F_{ms}(x))$$

$$x_{correct} = x + \Delta$$

Here, x is the daily maximum temperature; F is the cumulative distribution function (CDF) and F^{-1} is the inverse CDF; oc denotes observations in the training period; mc denotes model outputs in the training period; ms denotes model outputs in a validation period.

The Taylor diagram is used to assess the simulation capability of GCMs for maximum temperature in metropolises of China by applying the correlation coefficient RMSE and the standard deviation between the GCM output and the observation. We found that the correlation coefficient between the GCM output and the observational field can pass the significance test. The RMSE of most models is around 0.5, and their distributions are concentrated in the Taylor diagram. Results of the Taylor diagram show that the GCMs have good consistency in the simulation of the maximum temperature in China's major cities (Fig. 2).

Fig. 2 Taylor diagram of the maximum temperature in major cities of China between GCMs and observations (1, CNRM-CM5 r1i1p1; 2, CSIRO-Mk3-6-0 r10i1p1; 3, CSIRO-Mk3-6-0 r2i1p1; 4, CSIRO-Mk3-6-0 r3i1p1; 5, CSIRO-Mk3-6-0 r4i1p1; 6, CSIRO-Mk3-6-0 r5i1p1; 7, CSIRO-Mk3-6-0 r6i1p1; 8, CSIRO-Mk3-6-0 r7i1p1; 9, CSIRO-Mk3-6-0 r8i1p1; 10, CSIRO-Mk3-6-0 r9i1p1; 11, CanESM2 r1i1p1; 12, CanESM2 r2i1p1; 13, CanESM2 r3i1p1; 14, CanESM2 r4i1p1; 15, CanESM2 r5i1p1; 16, GFDL-CM3 r1i1p1; 17, GFDL-ESM2G r1i1p1; 18, GFDL-ESM2M r1i1p1; 19, HadGEM2-ES r1i1p1; 20, IPSL-CM5A-LR r1i1p1; 21, MIROC-ESM-CHEM r1i1p1; 22, MIROC-ESM r1i1p1; 23, MIROC5 r1i1p1; 24, MIROC5 r2i1p1; 25, MIROC5 r3i1p1; 26, MPI-ESM-LR r1i1p1; 27, MPI-ESM-LR r2i1p1; 28, MPI-ESM-LR r3i1p1; 29, MPI-ESM-MR r1i1p1; 30, MRI-CGCM3 r1i1p1; 31, NorESM1-M r1i1p1)

6. Introduction: I suggest a complete rewrite. Paragraph 2 doesn't say much and can be dropped. Paragraph 3 suggests that SSP and sex/age specificity will be the primary contribution of this study. Paragraph 4 talks about urbanization. Paragraph 5 talks about Paris targets. This is wide-ranging and lacks focus, and given the brevity of each paragraph the introduction really doesn't do justice to any of the topics. If I were writing this paper I would move Paris targets to Paragraph 2. One of the most compelling motivation for this study (to my read) is to understand the implications of 1.5 vs. 2.0 stabilization targets—there is a serious, policy-relevant result on this topic that the paper provides. I would then follow the over-arching Paris target motivation with the recognition that impacts will vary across groups and by geography, which motivates the sex/age/north-south aspects of the study. Personally, I don't think you even need to get into urban effects here, as nobody can argue that it's not important to look at health in megacities of China. Obviously, this is just one suggestion. My main concern is that the introduction should clearly motivate the top analyses performed in the paper. Right now it's more like a listing of important climate issues, and much of the results section that follows is similarly broad and hard to catch hold of.

Answer:

In the revised manuscript, the comments of the reviewer were fully considered. We made major revisions to the Introduction section, which now includes four parts: impacts of high-temperature on human health, review of research progress on heat-related morbidity and mortality, shortcomings of previous studies, and what is intended to be done in this article. A misleading content, i.e. the urban effects, is removed (P. 2-3, L. 30-86).

7. Results: I acknowledge that style of results presentations varies across fields, and that health literature leans towards large numbers of confidence intervals presented in text and extensive tables where simple-minded climate scientists like me want to see results boiled down into graphs and summary tables. But since this paper is being submitted to Nature Communications as a high-impact climate and health paper, I do think that some structure and distillation of results is necessary. I would ask the authors to remove lists of results that do not yield interesting differences and that are not interpreted in a meaningful way in the Discussion section. Instead, focus on the results that do yield insights, make sure that they are supported with appropriate statistical tests, and, where possible, highlight them in a figure or table on key findings. I personally get completely bogged down in these long lists of results, many of which appear to be statistically insignificant.

Answer:

According to the reviewer's comments, we removed several numbers that were of less importance and now focus our conclusions only on the numbers and figures that deliver meaningful information (see P. 7-8, L. 222-274).

Additional specific comments

1. I hate to harp on grammar, but the manuscript is difficult to follow in some passages. The authors or the journal editorial experts should attempt to edit grammar for clarity. As one example, the authors repeatedly employ the phrase "might be" when talking about projections. I'm not sure what is implied by that choice of words. I would opt for more precise language, such as "is projected to be between," or some similar phrasing. There are other examples like this throughout the text.

Answer:

We modified the grammar and syntax throughout the entire manuscript.

2. Line 110: This aside about definition is distracting. Either put it at the beginning of the presentation of results or let the reader find it in the methods section.

Answer:

In the revision, contents regarding heat wave have been deleted.

3. Line 117: Why aren't the 2006-2015 mortality results presented? The authors list the figures for morbidity in the previous paragraph but do not list the corresponding numbers for

mortality here.

Answer:

In the revised version, the projected morbidity and mortality data are compared with the reference period of 1986-2005, not with the period of 2006-2015.

4. Figure 2: This figure seems unnecessary. The four numbers that go into the large bar chart are already stated in the text. It's also not clear why the results from 2011-2013 are plotted, when much longer periods are discussed in the text. More broadly, I think this figure could be usefully replaced by a more informative table that summarizes the many 1986-2005 vs. 2006-2015 numbers on heat wave duration, intensity, and frequency that are listed in the text. A table would help the reader keep these numbers organized in his/her mind.

Answer:

A longer time series of the morbidity and mortality data for different gender and age groups in major cities of China could be developed (from originally 2011-2013 to 2007-2013 in the present, revised, draft). Additionally, changes in morbidity and mortality with global warming have been displayed in the form of figures (Figs 2-4 in main text) and tables (supplementary Table 1 and Table 3-5).

5. Figure 3: This figure should be supported by statistical tests of the trends the authors describe in the text. Are any of the trends significant?

Answer:

Figure 3 in the original manuscript is a description of past morbidity and mortality induced by heat-related events in China in 1961-2015, which has been replaced now by Fig. 1 in the revised version to show how high-temperatures are projected to change in future relative to the past. The significances of all trends in the paper are examined by applying the Mann-Kendall test.

6. Figure 4: The authors should provide appropriate statistical significance tests on the changes summarized in this figure and listed in the text throughout the section. Any time that North and South are contrasted in a quantitative way or a claim is made about present day vs. 1.5C warming vs. 2.0C warming a statistical test should really be provided.

Answer:

In the manuscript, trends in frequency and intensity of high-temperatures in 1961-2005 and in 2006-2100 are examined by applying the Mann-Kendall test (P. 3-4, L. 113-125).

7. Line 168: To the reader unfamiliar with SSP, this choice is entirely mysterious. How did the authors decide which SSP to associate with which RCP? I would suggest a paragraph or two on the SSP in the methods or supplementary materials, as I think that many readers interested in this study will not be familiar with the SSP.

Answer:

The Shared Socioeconomic Pathways (SSPs) describe a set of plausible alternative futures of societal development over the 21st century. The SSPs include a sustainable world (SSP1), a pathway of a continuing historical trend (SSP2), a strongly fragmented world (SSP3), a highly unequal world (SSP4), and a growth-oriented world (SSP5).

In this study, the projected population by age and gender, and the Gross Domestic Product (GDP) under all five SSPs have been presented for 27 major cities of China (see Supplementary SI-5). The combination of five SSPs with RCP2.6 and RCP4.5 represent ten climatic-socioeconomic conditions that form the basis for the assessment of heat-related morbidity and mortality in future.

Reviewers' comments:

Reviewer #1 (Remarks to the Author):

Overall Comments

The authors should be commended for their substantial effort in revising this manuscript and in-depth re-analyses using the state-of-the-art epidemiological and climate change projection approaches. In particular, they have reanalyzed the exposure-response functions using the DLNM and considered adaptation in projecting future climate change-related heat impacts. However, there are still several significant issues that should be addressed before being published in *Natural Communications*.

Specific Comments

1. The authors stated that they used 2060-2099 under RCP2.6 for 1.5 °C and under RCP4.5 for 2.0 °C. Is this 40-year average temperature used in determining the 1.5 °C and 2.0 °C scenarios, although the historical period is 20-years average (1986-2005)? In lines 124-126, the authors mentioned that “compared with the reference periods, the intensity of high temperature is projected to increase by 1.2 °C and 2.0 °C at global warming of 1.5 °C and 2.0 °C, respectively”. Does this mean that in the 1.5 °C scenario, the actual temperature increase is 1.2 °C instead of 1.5 °C? More clarifications in the future period used in the 1.5 °C and 2.0 °C scenarios should be given. For example, an overview table should be included to introduce the next period, global average temperature, and SSP population under the 1.5 °C and 2.0 °C scenarios

2. The biggest problem in estimating the heat-related morbidity is the use of an annual number of hospitalizations at the provincial level in time-series analysis. The use of annual rather than the real daily number of hospitalizations resulted in unreliable estimates. In temperature-health time-series studies, it is the daily variation that drives the health outcome contrast for temperature exposure. The temporal variation pattern between mortality and morbidity might be different, especially considering the seasonal change and day-of-the-week variation. Besides, using a proportion of the city population to the provincial total to get the city-level daily hospitalization data is also problematic in time-series study. The exposure-response function should be derived from the empirical time-series study, not modeling study as was conducted in this study. Therefore, I strongly recommend the authors to delete the morbidity analysis from this paper.

3. For gender, age, and GDP-specific SSP, it seems that the authors used national-level projections rather than for each city. This should be acknowledged as a limitation as the future trend may be different across regions (e.g., north vs. south; east vs. west) and cities.

4. The current title “Tens of thousands of additional deaths in major cities of China between the 1.5 °C and 2.0 °C warming” is inappropriate and misleading. For example, it is not clear to the readers whether this “tens of thousands of additional deaths” is per year, per decade, or during the 2060-2099 period.

Reviewer #2 (Remarks to the Author):

The authors changed statistical models to dlnm which is more complicated than previous piece-wise linear regression. And they extended study periods to 2007-2013. With this extended data, they had more stable temperature-mortality relationships.

I have concerns on morbidity data. The morbidity data were based on several assumptions. In the response the authors replied:

“Morbidity data in this study are based on hospitalization records. Annual number of hospitalizations in each province was recorded in the China Statistical Yearbook. The number of hospital admission in each city is deduced based on the proportion of the city population to the provincial total. We allocate the annual hospital admission into the daily scale, by assuming that the percentage of mortality to morbidity remains unchanged within a year (P. 9, L. 296-300).”

Figures 3 and 4 in the supplementary information are very similar (almost identical!) because of these assumptions. This is a very unrealistic assumption. I don't think reporting results about morbidity is acceptable if their calculations were based on Figures 3 and 4.

Reviewer #3 (Remarks to the Author):

I appreciate the careful attention the authors have given to reviewer comments. In my opinion the current manuscript is much improved, and I recommend it for publication.

Responses to the Reviewers' comments

Reviewer #1 (Remarks to the Author):

The authors should be commended for their substantial effort in revising this manuscript and in-depth re-analyses using the state-of-the-art epidemiological and climate change projection approaches. In particular, they have reanalyzed the exposure-response functions using the DLNM and considered adaptation in projecting future climate change-related heat impacts. However, there are still several significant issues that should be addressed before being published in Natural Communications.

1. The authors stated that they used 2060-2099 under RCP2.6 for 1.5 °C and under RCP4.5 for 2.0 °C. Is this 40-year average temperature used in determining the 1.5 °C and 2.0 °C scenarios, although the historical period is 20-years average (1986-2005)? In lines 124-126, the authors mentioned that “compared with the reference periods, the intensity of high temperature is projected to increase by 1.2 °C and 2.0 °C at global warming of 1.5 °C and 2.0 °C, respectively”. Does this mean that in the 1.5 °C scenario, the actual temperature increase is 1.2 °C instead of 1.5 °C? More clarifications in the future period used in the 1.5 °C and 2.0 °C scenarios should be given. For example, an overview table should be included to introduce the next period, global average temperature, and SSP population under the 1.5 °C and 2.0 °C scenarios

Answer:

(1) Our study applies 31 GCM outputs from the Coupled Model Intercomparison Project phase 5 (CMIP5) to deduce 1.5°C and 2.0°C global warming scenarios by using 20-year moving-average temperature. The multi-model ensemble mean shows that global mean temperature might reach the 1.5°C warming threshold (above the pre-industrial level) around 2020 - 2039 under the RCP2.6 scenario, and 2.0°C around 2040 - 2059 under RCP4.5. The projected temperature shows a low variation after the 2060s under both pathways. In order to conduct an impact study under comparative stable climatic conditions, we choose the common future time horizon of 2060 - 2099, corresponding to a 1.5°C global warming under RCP2.6 and a 2.0°C global warming under RCP4.5, although there will be overshoot. To make our study comparable with other impact studies, the reference interval 1986 - 2005 was chosen, consistently with many impact studies and the most recent IPCC reports (P. 3, L. 82-92.).

(2) The intensity of high-temperature is defined as the temperature above the mortality-inducing threshold temperature. In 1986 - 2005, the intensity of high-temperature averaged over 27 major cities in China was 1.6°C based on the ensemble mean of multiple GCMs, and it is projected to increase to 2.8°C and 3.5°C, respectively, at a global warming of 1.5°C and 2.0°C. That is to say, the intensity of high-temperature in China metropolises will increase by 1.2°C and 1.9°C at a global warming of 1.5°C and 2.0°C relative to 1986-2005, while the increase in global mean temperature is likely to be 0.89 °C and 1.39 °C at the same time (P. 4, L. 116-121).

(3) To clarify, Tables 4-6 are added in the Supplementary information that summarize socioeconomic and mortality changes in major cities of China.

2. The biggest problem in estimating the heat-related morbidity is the use of an annual number of hospitalizations at the provincial level in time-series analysis. The use of annual rather than the real daily number of hospitalizations resulted in unreliable estimates. In temperature-health time-series studies, it is the daily variation that drives the health outcome contrast for temperature exposure. The temporal variation pattern between mortality and morbidity might be different, especially considering the seasonal change and day-of-the-week variation. Besides, using a proportion of the city population to the provincial total to get the city-level daily hospitalization data is also problematic in time-series study. The exposure-response function should be derived from the empirical time-series study, not modeling study as was conducted in this study. Therefore, I strongly recommend the authors to delete the morbidity analysis from this paper.

Answer:

In the original paper, the number of hospital admissions in each city was deduced by double disaggregation, taking the proportion of the city population to the provincial total, and then temporally downscaling the annual number of hospital admissions into daily scale by assuming that the percentage of mortality to morbidity remains similar within a year.

To date, only one time series of daily hospitalization records is available to us. It stems from the emergency department of the Xinhai Hospital in Guangzhou city for 2009 - 2011, which was established in December 1981 and serves as comprehensive national second-class hospital.

We compared morbidity information given by the number of emergency cases recorded in the Xinhai Hospital and the estimated result from the disaggregation of annual hospitalizations in June, July, and August. Although the temporal trends in the two datasets are similar, the estimated daily morbidity in Guangzhou does not match the Xinhai Hospital record sufficiently well. Although both of them reflect higher morbidity in 2010 than in 2009 and 2011 (Fig.1), we have to admit that the data validation for Guangzhou alone cannot be considered as sufficiently robust to justify applicability in our study of 27 cities, and hence cannot reliably support the conclusions of the original article. Therefore, we follow the constructive advice of reviewers #1 and #2 to remove all the morbidity-related contents.

Fig.1 Comparison of the number of daily emergency admissions from Xinhai Hospital and the estimated daily morbidity of Guangzhou

3. For gender, age, and GDP-specific SSP, it seems that the authors used national-level projections rather than for each city. This should be acknowledged as a limitation as the future trend may be different across regions (e.g., north vs. south; east vs. west) and cities.

Answer:

The projected population and GDP under all five SSPs are available at the country level from the Organization for Economic Co-operation and Development (OECD) for the 21st century.

In our study, population and GDP for the 21st century are projected under the SSPs framework at the provincial scale, considering past and future socioeconomic development patterns, which were quantified by using regionally different parameters in the Population-Development-Environment model and the Cobb-Douglas production model (Leimbach et al., 2017; Huang et al., 2019 in the list of references). We have downscaled the province-scale population and GDP data into 0.5° resolution according to the county-level distribution of population and GDP in 2010. Finally, we extracted the city-level population and GDP data based on the present boundaries of each city. Although we do not project the socioeconomic development for each city, our data can still reflect considerable regional differences.

4. The current title “Tens of thousands of additional deaths in major cities of China between the 1.5 °C and 2.0 °C warming” is inappropriate and misleading. For example, it is not clear to the readers whether this “tens of thousands of additional deaths” is per year, per decade, or during the 2060-2099 period.

Answer:

We agree that the title should contain a temporal specification (per year or annually). Hence, we changed the title of the manuscript to “Tens of thousands of additional deaths annually in China cities between the 1.5 °C and 2.0 °C global warming”.

Reviewer #2 (Remarks to the Author):

The authors changed statistical models to dlnm which is more complicated than previous piece-wise linear regression. And they extended study periods to 2007-2013. With this extended data, they had more stable temperature-mortality relationships. I have concerns on morbidity data. The morbidity data were based on several assumptions. In the response the authors replied: "Morbidity data in this study are based on hospitalization records. Annual number of hospitalizations in each province was recorded in the China Statistical Yearbook. The number of hospital admission in each city is deduced based on the proportion of the city population to the provincial total. We allocate the annual hospital admission into the daily scale, by assuming that the percentage of mortality to morbidity remains unchanged within a year (P. 9, L. 296-300)." Figures 3 and 4 in the supplementary information are very similar (almost identical!) because of these assumptions. This is a very unrealistic assumption. I don't think reporting results about morbidity is acceptable if their calculations were based on Figures 3 and 4.

Answer:

Here, the same answer as to comment 2 by reviewer #1 applies. As suggested by both reviewers, we decided to remove all the morbidity-related contents within the manuscript.

REVIEWERS' COMMENTS:

Reviewer #1 (Remarks to the Author):

Many thanks to the authors' exceptional efforts to improve the manuscript and address my previous concerns satisfactorily. I recommend this paper to be published in Nature Communications.

Reviewer #2 (Remarks to the Author):

The authors deleted analyses of morbidity as I stated in the previous reviewed. The quality of the manuscript has been improved after revisions. I think it's ready to be published.